# Applying machine learning and geolocation techniques to social media data (Twitter) to develop a resource for urban planning

Sveta Milusheva[1]*, Robert Marty[1], Guadalupe Bedoya[1], Sarah Williams[2], Elizabeth Resor[3], Arianna Legovini[1]

1 Development Impact Evaluation Department, World Bank, Washington, DC, United States of America, 2 School of Architecture and Planning, Massachusetts Institute of Technology, Cambridge, MA, United States of America, 3 School of Information, University of California, Berkeley, CA, United States of America

* smilusheva@worldbank.org

**Data Availability Statement:** The twitter data and crash verification data used in the study are available via the World Bank's microdata catalogue. Please see here for the links to access the data: https://microdata.worldbank.org/index.php/catalog/

## Abstract

With all the recent attention focused on big data, it is easy to overlook that basic vital statistics remain difficult to obtain in most of the world. What makes this frustrating is that private companies hold potentially useful data, but it is not accessible by the people who can use it to track poverty, reduce disease, or build urban infrastructure. This project set out to test whether we can transform an openly available dataset (Twitter) into a resource for urban planning and development. We test our hypothesis by creating road traffic crash location data, which is scarce in most resource-poor environments but essential for addressing the number one cause of mortality for children over five and young adults. The research project scraped 874,588 traffic related tweets in Nairobi, Kenya, applied a machine learning model to capture the occurrence of a crash, and developed an improved geoparsing algorithm to identify its location. We geolocate 32,991 crash reports in Twitter for 2012–2020 and cluster them into 22,872 unique crashes during this period. For a subset of crashes reported on Twitter, a motorcycle delivery service was dispatched in real-time to verify the crash and its location; the results show 92% accuracy. To our knowledge this is the first geolocated dataset of crashes for the city and allowed us to produce the first crash map for Nairobi. Using a spatial clustering algorithm, we are able to locate portions of the road network (<1%) where 50% of the crashes identified occurred. Even with limitations in the representativeness of the data, the results can provide urban planners with useful information that can be used to target road safety improvements where resources are limited. The work shows how twitter data might be used to create other types of essential data for urban planning in resource poor environments.

## Introduction

The World Bank has declared that data is the next deprivation to end; they argue that the lack of data causes many of the world's poorest populations to be overlooked when resources are

3820 and https://microdata.worldbank.org/index.
php/catalog/3821.

**Funding:** • Initials of the authors who received each
award: (1) SM,GB,SW,AL; (2) SM,GB,SW,AL •
Grant numbers awarded to each author: N/A • The
full name of each funder: (1) UK Foreign,
Commonwealth & Development Office and (2) the
World Bank's Knowledge for Change Program •
URL of each funder website: (1) https://www.
gov.uk/government/organisations/
foreigncommonwealth-development-office; (2)
https://www.worldbank.org/en/programs/
knowledge-for-change.

**Competing interests:** The authors have declared
that no competing interests exist.

allocated to address their essential needs [1]. Data deprivation is a pressing challenge with as
many as 74% of the global and 97% of the Sub-Saharan African population living in countries
without adequate vital registration [2]; one third of countries lacking any poverty statistics [1];
and only 17% of the estimated road traffic deaths reported in official figures of low-income
countries [3]. Without data to inform national and urban policies, the gap between low- and
high-income countries will worsen [4]. However, while official statistics are poor, data in the
hands of private providers is plentiful, populated by the rapid expansion of mobile phones and
social media. Globally, phone penetration reached 67% in 2019 [5], and social media penetra-
tion is almost 50% [6]. This provides an opportunity for using crowdsourced data to study
major urban and development policies [7–11].

In this project we test the hypothesis of whether privately maintained data can be trans-
formed into a resource to better understand development challenges. Private data has been
used to characterize populations from determining poverty to understanding public emotions
[12–17]. Here, we use private data to describe the urban environment that affects those popu-
lations, specifically analyzing events reported on social media that affect people's safety such as
road traffic crashes, crime or floods. We focus on road traffic crashes (RTCs). Despite being
the number one cause of death for children and young adults aged 5-29 years, the lack of ade-
quate data on RTCs is a recognized and unmet challenge [18]. The objective is to improve
RTC data for urban planners so they can contribute to addressing the high toll of road deaths,
estimated globally at 1.35 million a year [3]. Our case study is Kenya, a country with high road
mortality, where the official figures are said to underestimate the number of fatalities by a fac-
tor of 4.5 [3].

The United Nations' Sustainable Development Goal (SDG) 3 sets a target to halve road
mortality by 2020; progress has been slow, and the target moved to 2030. The Stockholm
Declaration by the Third Global Ministerial Conference on Road Safety "Achieving Global
Goals 2030" reiterated the call for country investments in road safety–from legislation and
regulation, safe urban and transport design, safe modes of transport and vehicles, to modern
technologies for crash prevention, trauma care, and urban management. However, resource
constraints make it unlikely that countries will be able to meet all of these goals. Instead, coun-
tries should strategically invest for the greatest impact. This requires knowing where and when
crashes happen, so that resources can be targeted to risky locations and times.

Social media data, with all its biases, can contribute to filling some of the data gaps for
urban analysis, planning and management [19]. In this study, we create an algorithm that clas-
sifies transport-related tweets into geolocated RTCs for Nairobi. This is done by building on
existing literature to test two natural language processing algorithms to identify crash reports
[20, 21], developing an improved geoparsing algorithm to extract data on crash time and loca-
tion [22–28], and ground truthing the results. The paper also contributes to a broader litera-
ture that uses machine learning methods for road safety analysis [29–31].

This study innovates on three fronts and demonstrates the value of using social media to
expand data availability. (1) Geospatial Twitter data analysis usually uses the approximately
1% of tweets that have a geolocation tag [32–34]; we improve this by using a machine learning
geoparsing algorithm to leverage the 99% of tweets that do not contain a geotag. (2) To our
knowledge there are no other studies that physically validate the locational accuracy of tweets
in real time. 92% of verified tweets were found to be valid crashes, demonstrating the validity
of crowdsourced crash data. (3) The work created an essential resource by generating one of
the the first real-time maps of RTCs in an African city (Nairobi). We identify 52,228 crash
reports and geolocate those with enough information provided in the text (32,991 of them).
In a context where there is no systematic georeferenced data on crashes to support policy

planning, the process outlined here could be used to capture this data for cities all over the world that need this essential resource.

Overall, the method expands the coverage of road crashes that can be used to analyze road safety and to prioritize policy action around the locations where crashes occur more often. This is especially useful in country contexts where the only data available for analysis are aggregated statistics on total fatalities in the country, with no detailed breakdown of location or time. Crowdsourced data can help act as an additional input that can be used by policymakers in better understanding the situation. By using a clustering algorithm to identify and rank crash locations, we find that the top 15% of crash clusters (66 out of 435) account for half of all crashes. Knowing that a small portion (<1%) of the road network hosts 50% of RTCs in the crowdsourced data can help reduce an intractable problem to a more manageable one. This analysis shows the potential for using this data to complement road safety diagnostics and to guide investments and planning in road safety in Kenya and in other contexts, especially those with similar data defi-ciencies and with sufficient social media density like India and the Philippines [35].

The approach can be extended to other events reported on social media, whether related to disaster relief, crime, personal safety, urban mobility, or road maintenance. The work on disas-ter relief and response makes prominent use of geoparsing of tweets [36–43]. Geoparsing of tweets that lack geolocation information could enable more comprehensive crime analytics [44–46]. Improved algorithms can lead to faster and better geolocation of events, which would help urban planners and policy makers improve responses and better target interventions.

## Method

The research for this study was approved by the Committee on the Use of Humans as Experi-mental Subjects (COUHES; COUHES protocol #1711128913). Consent was not obtained from participants. The research involved observation of motor vehicle crashes and recording the observable human and property damages from these crashes; only features observable from a distance were captured and no identifiable information was collected. Secondary data used were not collected specifically for this research and any identifiable information was removed prior to analysis.

The goals of this analysis are to create data on road crashes with times and locations and understand how these incidents cluster in the city, which allows for the spatial prioritization of urban investments in road safety. The technical challenges this study addresses are: i) improve the protocols for geolocation, ii) apply applications of AI to classify tweets reporting crashes and identify their location from multiple geographical references, iii) cluster the crashes geo-graphically and identify areas with many crashes. See S1 File for detailed methodology. The components are as follows:

1. **Scrape data**. We scrape 874,588 tweets posted by Ma3Route, an existing urban mobility platform with 1.1 million followers, since its inception in May 2012 through July 2020 (see S1 File for examples of tweets and for a figure of the daily number of tweets across time). We scrape tweets in compliance with Twitter's Terms of Service using the premium Search Tweets Full Archive API.

2. **Develop and augment a gazetteer**. We build a gazetteer of landmarks for the five counties that constitute the Nairobi metro area using: OpenStreetMap, Geonames and Google Places. The gazetteer includes the landmark name, geocoordinates and type of landmark (e.g., school, bus stop). We use consecutive combinations of 2 and 3 words (known as n-grams) and skip-grams of landmarks in the gazetteer, alternate spellings and abbreviations, and splitting of landmarks with select punctuation (e.g., slashes, parentheses, commas). We

innovate by developing alternate names that exclude the landmark type from the name (e.g., excluding "Hotel" from the name).

3. **Develop a truth dataset**. We develop a truth dataset to train the algorithm. Taking all tweets for July 2017—July 2018, we restrict tweets to the ones most likely related to a crash based on a broad list of words and their variations. Each tweet is manually coded, indicating (1) if the tweet reported a crash and (2) the approximate latitude and longitude of any reported crash whenever enough information is provided. 9,480 tweets were coded, of which 69% (6,602) reported a crash and of these, 63% (4,192) identified an approximate location of the crash. On average, users posted 10 crash reports that could be geolocated to Twitter daily.

4. **Identify RTCs and their location**. We use a three-step process to convert unstructured crowdsourced text into a dataset. The first is to identify relevant reports from hundreds of thousands of reports. The second is to extract necessary information from the relevant reports. The third is to consolidate unique record information from multiple reports of the same event. In Fig 1, we illustrate how the algorithm works to classify and geolocate RTCs. We use the tweet "Bad accident on Waiyaki Way next to Kianda heading towards ABC Place."

(a). **Classify relevant crowdsourced reports**. We restrict the analysis to tweets that contain keywords from a broad list of English and Kiswahili road safety terms such as "accident" or "overturn." This approach follows previous research and allows for misspellings [20]. We use natural language processing to classify and exclude tweets that contain road safety keywords but discuss road safety conditions rather than specific crash events (e.g., "terrible drivers keep causing crashes"). We test two approaches that analyze the combination of words in a tweet: Naive Bayes and support vector machines (SVM).

(b). **Geolocate reports**. We extract all landmarks and roads that have an exact match between the gazetteer and the tweet. In Fig 1, "kianda" and "abc way" match several entries in the gazetteer. We extract misspelled matches based on Levenshtein distance varied by length of the n-gram, matches based on the word following a preposition, and matches based on intersections between multiple roads.

Existing geoparsers extract all possible location references without identifying the unique location that makes the data useful. We resolve two technical challenges to find the location of the crash:

i. When multiple locations are mentioned in the tweets, we use prepositions to sort locations into tiers, based on the probability of a location being correct after a particular preposition. For example, in Fig 1, "next to" is ranked as tier 1 while "toward" is ranked as tier 6, resulting in the correct geolocation for the crash at "kianda" and not "abc place".

ii. When a name refers to multiple landmarks, we adopt a toponym resolution approach. In Fig 1, more than 6 landmarks across Nairobi have "kianda" in the name. We resolve the toponym in three steps: (1) we search for landmarks that are within 500 m of a road if it is mentioned, (2) we use the centroid of the clustered location if 90% or more of the landmarks are in a 500 m radius, or (3) we rank the landmarks by the probability of being correct using the landmark type in the truth data (see S1 File for statistics on location type). In the example, we use "Waiyaki Way" to narrow down the landmarks "kianda" in a 500 m radius (from 6 to 3) and then use the centroid as the crash location.

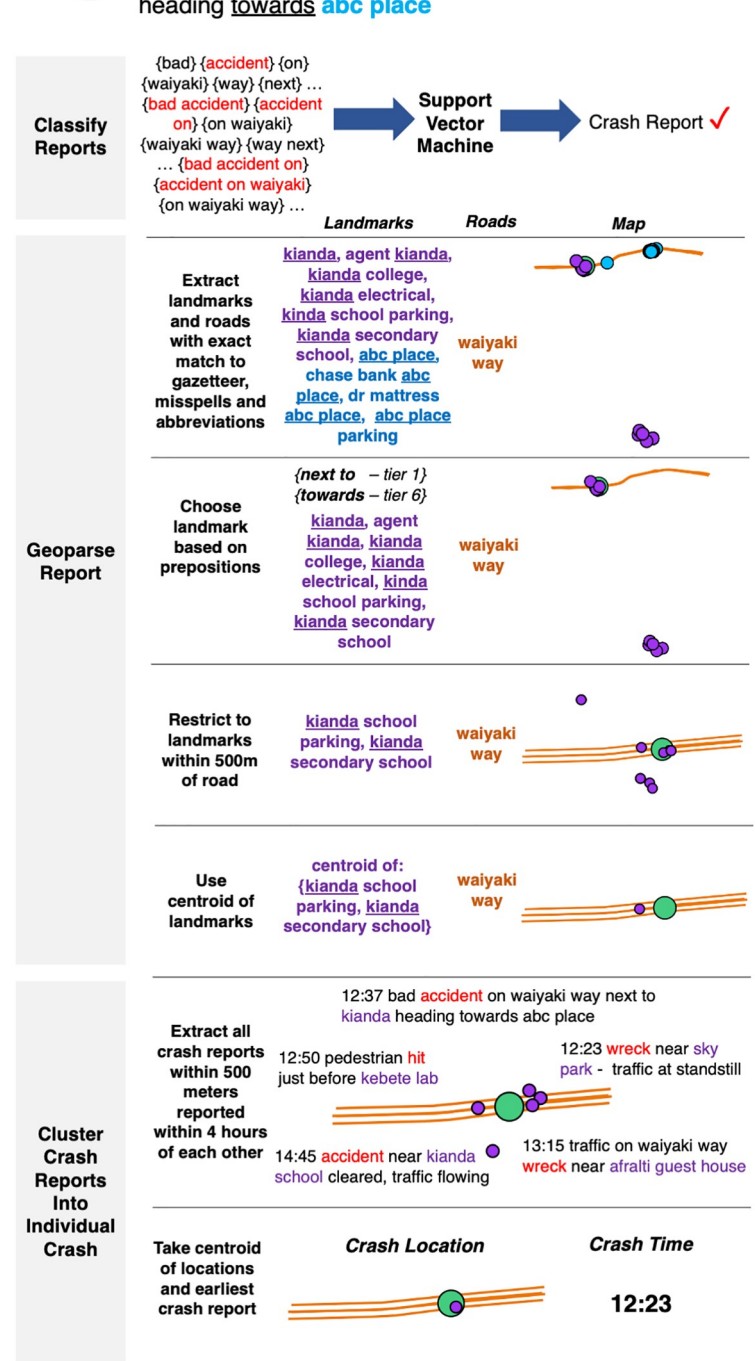

**Fig 1. Illustration of classification and geolocation algorithm developed for extracting data from crowdsourced information.**

We define a correct geoparse as one located within 500 m of the coordinates in the truth dataset. As a benchmark, we compare our algorithm to the Location Name Extraction tool (LNEx), which was shown to have better accuracy than other geoparsers [40]. As LNEx and other geoparsers are not designed to extract one unique location from text [26, 40, 47], we first judge performance by examining whether any location references are near the true coordinates. Next, we define the crash location as determined by LNEx to be the centroid of all locations it finds in the tweet and compare this with the unique location identified by our algorithm.

(c). **Identify unique reports**. To avoid over-counting, we develop a clustering algorithm that uses time and location to identify which tweets refer to the same crash. In Fig 1, five tweets report a crash within two hours of each other, referencing different landmarks that are all close together. To develop reasonable parameters for clustering, we manually identify tweets that report the same crash in the truth dataset based on the time, location and crash characteristics. The 4,192 crash reports are clustered into 2,648 unique crashes. For unique crash clusters, 97% of tweets reported landmarks within 500 m and within 4 hours of each other (see additional details in S1 File for how parameters were chosen).

(d). **Ground truth**. To ensure that the crowdsourced data is reliable and provides correct information, we conduct a ground-truthing exercise to validate the quality of the data and the performance of the underlying algorithm. We processed tweets in real-time and dispatched a motorcycle delivery service (Sendy) to the site of the crash within minutes. The Sendy driver was tasked with verifying and reporting whether a crash actually happened in that location. If a driver could not see the crash, they were instructed to ask a bystander whether a crash had occurred but was cleared or whether a crash occurred nearby. Drivers were able to arrive at the crash location quickly; the median time between being alerted of a crash and arriving at the scene was 26 minutes.

## Results

The methods laid out here created a georeferenced RTC dataset that was previously unattainable and produced one of the first real-time maps of RTCs in Nairobi. We classify 52,228 tweets as crash-related out of a universe of 874,588 tweets during 2012—2020 (Panel A of Fig 2). This is based on the SVM algorithm, which we find performs better than the Naive Bayes algorithm according to the F1 statistic (see S4 Table in the S1 File). We geolocate 32,991 time-stamped crash tweets from August 2012 to July 2020 and cluster them into 22,872 unique geo-located crashes (panels B and C of Fig 2 show the unique crashes generated by Twitter daily using the algorithm and clustering). In our truth dataset, where we manually coded each crash-related tweet, we found that 63% of tweets contain enough information in order to be geolocated. Assuming the same proportion of tweets contain enough information to be geolocated in the full dataset, we would expect 32,903 tweets with enough location information. This aligns almost perfectly with the number of tweets that the algorithm is able to geolocate.

The ground-truthing exercise confirms the validity of the crowdsourced data. We find that of the 73 crash-related tweets physically verified, 92% correctly corresponded to a crash near the estimated location; 32.8% witnessed the crash scene, 57.5% did not see the crash but were told by a bystander that a crash occurred and was recently cleared, and 1.4% reported that the crash did not occur at the specified location but nearby. Furthermore, using our truth dataset to benchmark shows that our algorithm performs significantly better than the current geoparsing standard. Our algorithm's recall rate of 65% is a five-fold improvement in performance compared to the LNEx algorithm (13% recall) in identifying the unique location of a crash

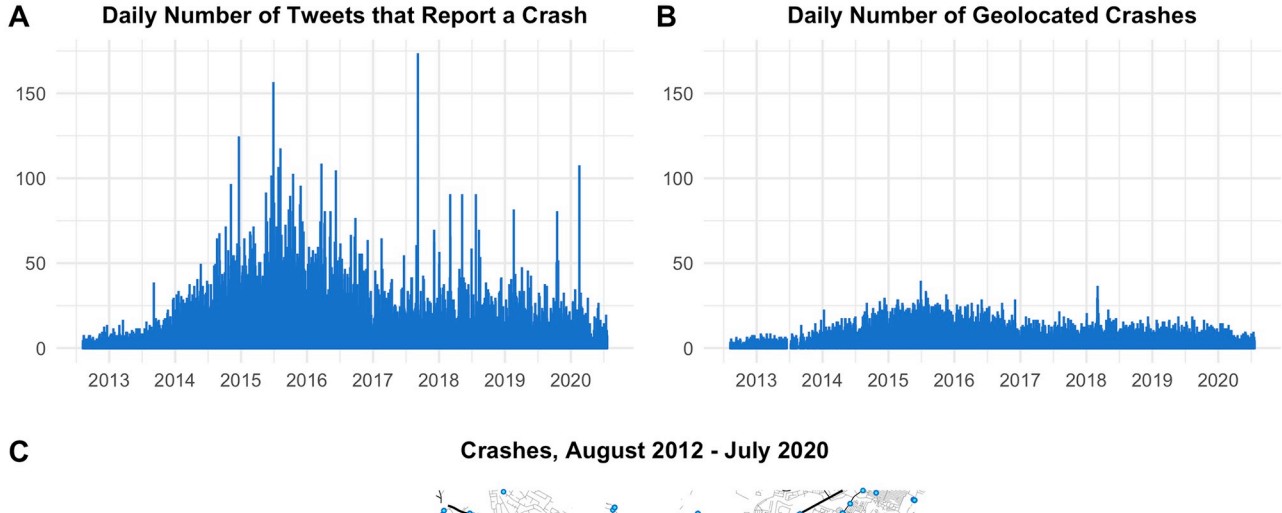

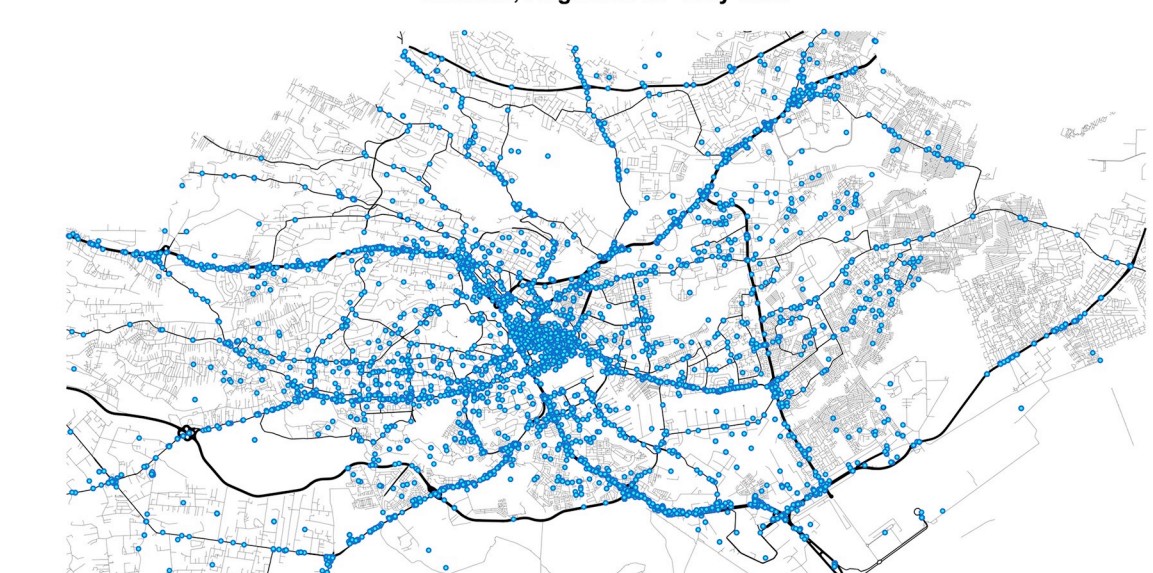

**Fig 2. Crowdsourced crash reports from twitter data that our algorithm has geolocated and clustered into unique crashes for the city of Nairobi between 2012 and 2020.** Road data comes from OpenStreetMap.

(Table 1). This is largely because LNEx is not designed to identify a unique location when multiple locations are mentioned. Our algorithm performs 25% better than LNEx even when comparing whether any location extracted from the tweet is near the true location.

Analyzing the crash data produced using our algorithm and focusing on the truth dataset within the city limits of Nairobi, we find that all crashes from July 2017 to July 2018 can be found in 435 clusters, each with a maximum diameter of 300 m. Of these clusters, 67% have two or more crashes and there are 56 clusters with 10 or more crashes. Additionally, 66 crash clusters represent over 50% of all the crashes. When looking at the 7.5 years of crowdsourced data for the city of Nairobi, the number of crash clusters do not grow linearly, implying that the locations where crashes occur and are reported in Twitter are consistent across years. Only 14% of crash locations have only a single crash, and there are 443 crash clusters with 10 or more crashes. We see the concentration of crashes even more when we note that only 9% of crash clusters (133 our of 1375) represent 50% of the crashes reported (Fig 3 shows crash heatmaps for the truth dataset from July 2017 to July 2018 and for 2012-2020).

**Table 1. Geolocation algorithm results.**

| | Any Location Captured by Algorithm Close to True Crash Location | | Crash Location Determined by Algorithm Close to True Crash Location | |
|---|---|---|---|---|
| | Recall | Precision | Recall | Precision |
| LNEx | 0.674 | 0.686 | 0.129 | 0.132 |
| Alg., Raw Gaz | 0.695 | 0.757 | 0.579 | 0.756 |
| Alg., Aug Gaz | 0.798 | 0.857 | 0.651 | 0.811 |
| Alg., Aug Gaz [Cluster] | | | 0.656 | 0.774 |

'N Crashes' refers to the number of correctly identified crashes. 'Raw Gaz' refers to the raw gazetteer (ie, dictionary of landmarks with original names) and 'Aug Gaz' refers to the augmented gazetteer. We use our raw gazetteer as an input into LNEX, which implements its own augmentation process. For LNEx, the crash location is determined by taking the centroid of all locations captured by the algorithm. Locations are considered close if they are within 500 meters of each other.

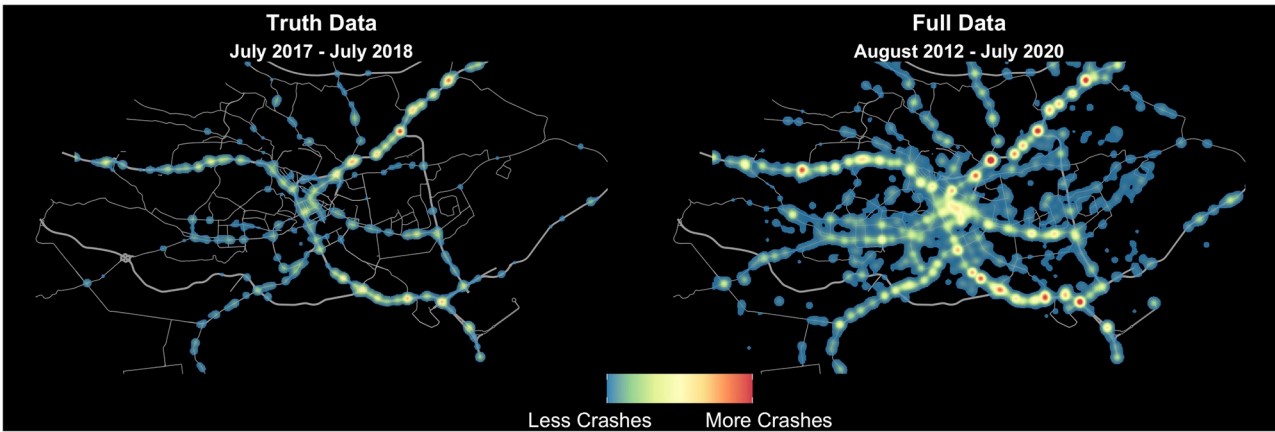

**Fig 3. Heatmap of crashes.** Data in panel a is from July 2017—July 2018, where we use the manually coded Twitter dataset. Data in panel b is for August 2012—July 2020. Road data comes from OpenStreetMap.

## Discussion

Cities are constantly evolving and understanding urban mobility is critical to creating urban designs that help to manage risks for pedestrians and vehicles. Severe data limitations hinder the development of policy interventions needed to manage risks, especially in low- and middle-income resource-constrained countries. Closing the data deprivation gap can help avert divergence in socioeconomic conditions between data-poor and -rich countries. By focusing on RTCs–the number one cause of death among young people—we demonstrate that social media could be an inexpensive way to produce non-existent RTC data in resource-poor contexts that can support government analyses of road safety and potentially inform policy. This tool could be especially powerful when combined with investments in building a digital administrative dataset that would provide information on the crashes attended by police. The answer to the seemingly simple question of where and when crashes occur has profound implications for public policy response that can save lives. And while official data deprivation can be an impediment to economic development, data generated by private operators can be transformed and placed in the hands of policy makers as a resource for policy making. By expanding the amount of data, we can generate more input to help resource-constrained countries prioritize policy action where it is most needed.

This example of geolocating crash data from mining twitter data can help to guide infrastructure redesign or enforcement policies to reduce RTCs. Nairobi comprises an extensive road network of 6200 km; with the city's limited resources, addressing road safety across the whole network is difficult. By using this type of geolocated data, urban planners and policy makers can narrow down the problem to the areas with the highest number of crashes. This has been proven to work in developed countries where targeting risky locations led to reductions in the concentration of crashes [48]. As shown in the results, crashes reported on Twitter are highly concentrated, with the top 15% of locations spread across 20 km of road having 50% of the crashes reported on Twitter.

It should be noted that there are some limitations to the approach. The data generated is limited by the coverage of the crowdsourced data. Users are more active on social media at particular times, and it is necessary to possess a smartphone and have access to internet to be able to use the service. This can lead to bias in the reports generated via the crowdsourced data. Only 7.5% of tweets are sent between the hours of 9 p.m. and 6 a.m., and as a result only 12% of the crash reports from Twitter are during this time. There could also be geographic bias if there are areas of the city where people with smartphones are more likely to be present or passing by, and therefore more likely to report. Our real-time motorcycle validation exercise demonstrates the internal validity of the crowdsourced data and the improved algorithm. External validity is more difficult to assess because we do not know what the universe of crashes is. Additionally, we do not know the severity of the crashes reported on Twitter. Therefore, we have no way of knowing if the areas where crashes happen are the most dangerous, which is what policy makers likely would want to target. These caveats should be considered by policy makers when using crowdsourced data to inform policies and targeting.

Despite the limitations, our improved geoparsing algorithm discussed in this paper can begin filling some of the gaps in data in low-capacity and data-scarce settings. And while the crash cluster areas identified by the algorithm may not be the most dangerous or may not represent all crash areas, they nevertheless highlight problem areas. All crashes, minor or severe, have important economic consequences in terms of property damage and lost time and productivity due to the traffic generated (which is one of the reasons the crash is likely reported on Twitter). Therefore, this data can be used to target areas for design solutions where we are seeing high numbers of crashes consistently over time. In settings where there are limited or non-existent administrative records and, therefore, lack of any geolocated data, this tool can produce information in real-time for one of the most pressing challenges in developing countries.

Furthermore, by developing tools that generate time-stamped geolocated data and statistics from crowdsourcing on different "events" that are reported on social media, we can hope to expand data availability across other contexts and across issues beyond RTCs. For example, real-time traffic applications like RIDLR in India can be used to expand data on road safety. These improved tools can also help geolocate victims during a natural disaster or alert disaster management teams to the location of unsafe buildings or areas needing immediate attention. They can support law-enforcement or communities to locate and respond to crimes, cases of violence against women, or police violence. Improved identification of time and location of events can help to automate and accelerate policy response across a wide set of issues, potentially leading to better policy outcomes.

## Supporting information

**S1 File.**
(PDF)

## Acknowledgments

We thank Robert Tenorio and Amy Dolinger for their field coordination and research support. We are also grateful to Andrew Muriithi, Purity Kimuru, Rodgers Avuya, Salome Omondi and Pheliciah Mwachofi for their field support. We appreciate comments from anonymous reviewers and participants at the ACM COMPASS Conference and the Netmob Conference.

## Author Contributions

**Conceptualization:** Sveta Milusheva, Guadalupe Bedoya, Sarah Williams, Elizabeth Resor, Arianna Legovini.

**Formal analysis:** Sveta Milusheva, Robert Marty, Guadalupe Bedoya.

**Investigation:** Sveta Milusheva, Robert Marty, Guadalupe Bedoya.

**Methodology:** Robert Marty, Sarah Williams, Elizabeth Resor.

**Writing – original draft:** Sveta Milusheva, Robert Marty, Guadalupe Bedoya, Sarah Williams, Arianna Legovini.

**Writing – review & editing:** Sveta Milusheva, Robert Marty, Guadalupe Bedoya, Sarah Williams, Arianna Legovini.

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
