## [Decision Letter · Decision Letter 0]

1 Oct 2020

PONE-D-20-27914

Applying machine learning and geolocation techniques to social media data (Twitter) to develop a resource for urban planning

PLOS ONE

Dear Dr. Williams,

Thank you for submitting your manuscript to PLOS ONE. After careful consideration, we feel that it has merit but does not fully meet PLOS ONE’s publication criteria as it currently stands. Therefore, we invite you to submit a revised version of the manuscript that addresses the points raised during the review process.

We look forward to receiving your revised manuscript.

Kind regards,

Feng Chen

Academic Editor

PLOS ONE

Journal Requirements:

3.We note that [Figure(s) 2 and 3] in your submission contain [map/satellite] images which may be copyrighted. All PLOS content is published under the Creative Commons Attribution License (CC BY 4.0), which means that the manuscript, images, and Supporting Information files will be freely available online, and any third party is permitted to access, download, copy, distribute, and use these materials in any way, even commercially, with proper attribution. For these reasons, we cannot publish previously copyrighted maps or satellite images created using proprietary data, such as Google software (Google Maps, Street View, and Earth). For more information, see our copyright guidelines: http://journals.plos.org/plosone/s/licenses-and-copyright.

1.    You may seek permission from the original copyright holder of Figure(s) [2 and 3] to publish the content specifically under the CC BY 4.0 license. 

Reviewers' comments:

Reviewer's Responses to Questions

**Comments to the Author**

1. Is the manuscript technically sound, and do the data support the conclusions?

Reviewer #1: Yes

Reviewer #2: Yes

2. Has the statistical analysis been performed appropriately and rigorously? 

Reviewer #1: Yes

Reviewer #2: Yes

3. Have the authors made all data underlying the findings in their manuscript fully available?

Reviewer #1: Yes

Reviewer #2: Yes

4. Is the manuscript presented in an intelligible fashion and written in standard English?

Reviewer #1: Yes

Reviewer #2: Yes

5. Review Comments to the Author

Reviewer #1: This paper applies machine learning and geolocation techniques to social media data (Twitter) to develop a resource for road traffic crashes. The research topic is interesting and worth of investigation. The performance of the proposed methods is demonstrated by a case study conducted in Nairobi, Kenya. The paper is generally well organized and written. A minor suggestion is that more works on applying machine learning in road traffic crashes are acknowledged in the Introduction section, such as:

Rule extraction from an optimized neural network for traffic crash frequency modeling. Accident Analysis and Prevention, 2016, 97: 87-95.

Modeling nonlinear relationship between crash frequency by severity and contributing factors by neural networks. Analytic Methods in Accident Research, 2016, 10: 12-25.

Reviewer #2: The topic of this paper is interesting. The methods sound. The results are meaningful and useful. There are several suggestions to improve this paper.

1. “We identify 57,886 crash reports and geolocate 36,428 of them.”The percentage of geolocating the crash reports is low. The authors need to clarify this issue.

2. "92% of verified tweets correspond to a crash, demonstrating the validity of crowdsourced crash data." This findings are great. The authors could mention the instructions and future directions for verifying tweets and relating these verified tweets with crash data.

6. PLOS authors have the option to publish the peer review history of their article (what does this mean?). If published, this will include your full peer review and any attached files.

Reviewer #1: No

Reviewer #2: No

---

## [Author Response · Author response to Decision Letter 0]

23 Nov 2020

Please see below our response to each of the reviewers comments in turn. 

Reviewer #1: This paper applies machine learning and geolocation techniques to social media data (Twitter) to develop a resource for road traffic crashes. The research topic is interesting and worth of investigation. The performance of the proposed methods is demonstrated by a case study conducted in Nairobi, Kenya. The paper is generally well organized and written. A minor suggestion is that more works on applying machine learning in road traffic crashes are acknowledged in the Introduction section, such as:

Rule extraction from an optimized neural network for traffic crash frequency modeling. Accident Analysis and Prevention, 2016, 97: 87-95.

Modeling nonlinear relationship between crash frequency by severity and contributing factors by neural networks. Analytic Methods in Accident Research, 2016, 10: 12-25.

Response: Thank you for pointing out this gap in the literature review. We have added both references, which nicely demonstrate the use of machine learning more broadly for road safety analysis, in addition to the following paper:

A comparative study on machine learning based algorithms for prediction of motorcycle crash severity. PLoS One, 2019; 14(4)

Reviewer #2: The topic of this paper is interesting. The methods sound. The results are meaningful and useful. There are several suggestions to improve this paper.

1. “We identify 57,886 crash reports and geolocate 36,428 of them.” The percentage of geolocating the crash reports is low. The authors need to clarify this issue.

Response: Thank you for this comment. While 57,886 (now updated in the paper to 52,228) tweets are identified as crash reports, there is a large percentage of them where there is not enough information provided in the tweet to geolocate. In our Truth dataset where tweets were individually coded by a team of local enumerators familiar with the city and landmarks, we found that only 63% of tweets reporting a crash contain enough information to identify an approximate location. If we are to assume the same percentage of tweets contain enough information to be identified in the full dataset of 52,228 crash tweets, then there should be around 32,903 tweets with enough information to be geolocated. This aligns almost perfectly with the number of tweets that the algorithm was able to geolocate (updated in paper to 32,991). We agree that this was not clear in the text and we have added language to provide more clarity.

2. "92% of verified tweets correspond to a crash, demonstrating the validity of crowdsourced crash data." This findings are great. The authors could mention the instructions and future directions for verifying tweets and relating these verified tweets with crash data.

Response: I am glad you liked the method. We have included additional details in the text about the ground truthing process. The details now read:

The Sendy driver was tasked with verifying and reporting whether a crash actually happened in that location. If a driver could not see the crash, they were instructed to ask a bystander whether a crash had occurred but was cleared or whether a crash occurred nearby. Drivers were able to arrive at the crash location quickly; the median time between being alerted of a crash and arriving at the scene was 26 minutes.

---

## [Decision Letter · Decision Letter 1]

8 Dec 2020

Applying machine learning and geolocation techniques to social media data (Twitter) to develop a resource for urban planning

PONE-D-20-27914R1

Dear Dr. Williams,

We’re pleased to inform you that your manuscript has been judged scientifically suitable for publication and will be formally accepted for publication once it meets all outstanding technical requirements.

Kind regards,

Feng Chen

Academic Editor

PLOS ONE

Additional Editor Comments (optional):

Reviewers' comments:

Reviewer's Responses to Questions

**Comments to the Author**

1. If the authors have adequately addressed your comments raised in a previous round of review and you feel that this manuscript is now acceptable for publication, you may indicate that here to bypass the “Comments to the Author” section, enter your conflict of interest statement in the “Confidential to Editor” section, and submit your "Accept" recommendation.

Reviewer #1: All comments have been addressed

Reviewer #2: All comments have been addressed

2. Is the manuscript technically sound, and do the data support the conclusions?

Reviewer #1: (No Response)

Reviewer #2: Yes

3. Has the statistical analysis been performed appropriately and rigorously? 

Reviewer #1: (No Response)

Reviewer #2: Yes

4. Have the authors made all data underlying the findings in their manuscript fully available?

Reviewer #1: (No Response)

Reviewer #2: Yes

5. Is the manuscript presented in an intelligible fashion and written in standard English?

Reviewer #1: (No Response)

Reviewer #2: Yes

6. Review Comments to the Author

Reviewer #1: (No Response)

Reviewer #2: (No Response)

7. PLOS authors have the option to publish the peer review history of their article (what does this mean?). If published, this will include your full peer review and any attached files.

Reviewer #1: No

Reviewer #2: No

---

## [Editor Report · Acceptance letter]

16 Dec 2020

PONE-D-20-27914R1 

Applying machine learning and geolocation techniques to social media data (Twitter) to develop a resource for urban planning 

Dear Dr. Williams:

I'm pleased to inform you that your manuscript has been deemed suitable for publication in PLOS ONE. Congratulations! Your manuscript is now with our production department. 

Kind regards, 

on behalf of

Dr. Feng Chen 

Academic Editor

PLOS ONE